# A family of lead clusters with precious metal cores

Cong-Cong Shu 1,3, Harry W. T. Morgan 2,3, Lei Qiao 1, John E. McGrady 2✉ & Zhong-Ming Sun 1✉

Gold nanoparticles have been used for centuries, both for decoration and in medical applications. More recently, many of the major advances in cluster chemistry have involved well-defined clusters containing tens or hundreds of atoms, either with or without a ligand shell. In this paper we report the synthesis of two gold/lead clusters, $[Au_8Pb_{33}]^{6-}$ and $[Au_{12}Pb_{44}]^{8-}$, both of which contain *nido* $[Au@Pb_{11}]^{3-}$ icosahedra surrounding a core of Au atoms. Analogues of these large clusters are not found in the corresponding Ag chemistry: instead, the Ag-centered *nido* icosahedron, $[Ag@Pb_{11}]^{3-}$, is the only isolated product. The structural chemistry, along with the mass spectrometry which shows the existence of $[Au_2Pb_{11}]^{2-}$ but not $[Ag_2Pb_{11}]^{2-}$, leads us to propose that the former species is the key intermediate in the growth of the larger clusters. Density functional theory indicates that secondary π-type interactions between the $[Au@Pb_{11}]^{3-}$ ligands and the gold core play a significant part in stabilizing the larger clusters.

---

[1] Tianjin Key Lab for Rare Earth Materials and Applications, State Key Laboratory of Elemento-Organic Chemistry, School of Materials Science and Engineering, Nankai University, Tianjin 300350, China. [2] Department of Chemistry, University of Oxford, South Parks Road, Oxford OX1 3QR, UK. [3]These authors contributed equally: Cong-Cong Shu, Harry W. T. Morgan. ✉email: john.mcgrady@chem.ox.ac.uk; sunlab@nankai.edu.cn

Gold clusters have long held the attention of chemists, in part because of their often spectacular highly symmetric geometries but also because they have recently found applications in medicine and nanotechnology[1,2]. At one extreme are the naked clusters $[Au_x]^{+/0/-}$ which have been studied using a variety of gas-phase spectroscopies[3] while at the other, thiolate-supported clusters with ever-increasing dimensions continue to emerge. Amongst these, Kornberg's $Au_{102}(S-C_6H_4-COOH)_{44}$ and the recently reported $Au_{279}(S-C_6H_4-{}^tBu)_{84}$ "Faradaurate-279" are particularly striking examples of how a spherical core of gold atoms can be stabilized by di-, tri- and tetrameric anionic 'staple ligands', $RS(AuSR)_n$ ($n = 1, 2, 3$) which constitute a protective 'mantle' around the cluster[4,5]. Häkkinen and colleagues have coined the term 'divide and protect' to describe the way that the gold content is separated into a zero-valent $Au^0$ core and monovalent $Au^+$-containing thiolate 'staples', formally anionic 4-electron donor ligands that bind to two atoms of the core via lone pairs on the terminal sulfur atoms[6]. At the core of many of these clusters, it is possible to identify high-symmetry $Au_x$ units, perhaps the most prominent being the 8-electron $[Au_{13}]^{5+}$ icosahedron, found, for example, in $[Au_{25}(S-C_6H_4-COOH)_{18}]^-$ and also in phosphine-ligated systems such as $[Au_{13}Cl_2(PMe_2Ph)_{10}]^{3+}$[7,8]. Mingos' theoretical work has shown that the relationship between structure and electron count in these and other gold clusters can be understood in terms of overlap of radially directed $s/d_{z^2}$ hybrids on each gold atom[9,10]. This model accounts elegantly for the approximately spherical geometries of $[Au_4(P^tBu_3)_4]^{2+}$ ($1S^2$) and $[Au_{13}Cl_2(PMe_2Ph)_{10}]^{3+}$ ($1S^21P^6$) as well as the prolate and oblate distortions found in $[Au_6(PPh_3)_6]^{2+}$ ($1S^21P^2$) and $[Au_7(PPh_3)_7]^+$ ($1S^21P^4$), respectively[11–13]. Clusters with 8 gold atoms, in contrast, tend to adopt rather less symmetric structures, such as the 'core + exo' geometries of $[Au_8(dppp)_4]^{2+}$ and $[Au_8(dppp)_4X_2]^{2+}$, X=Cl−, PhC≡C− (Fig. 1) where an octahedral $Au_6$ core is capped by two 'exo' gold atoms[14,15], or the highly distorted cube reported recently for $[Au_8(PPh_3)_8]^{2+}$[13].

In contrast to the well-established chemistry of gold clusters with thiolate or phosphine ligands, there has been only one previous report of a Zintl-ion cluster containing gold, that being the approximately icosahedral 62-electron $[Au@Pb_{12}]^{3-}$ reported by some of us in 2017[16]. The wider family of endohedral lead clusters also includes 60-electron $[M@Pb_{12}]^{q-}$ (M=Ni, Pd, Pt, $q = 2$, M=Co, Rh, Ir, $q = 3$) and 58-electron $[Mn@Pb_{12}]^{3-}$[17–21]. The 60-electron count has 'magic' status in this family, and also in the analogous clusters of Sn, simply because it corresponds to closed-shell configurations at both M ($d^{10}$) and the closo $[Pb_{12}]^{2-}$ cage ($4n + 2 = 50$ where $n$ is the number of vertices), and indeed the empty $[Pb_{12}]^{2-}$ cage has itself been identified as a stable entity in the gas phase. Smaller lead cages are also known; for example homometallic $[Pb_{10}]^{2-}$ and its nickel-centered analogue $[Ni@Pb_{10}]^{2-}$ have both been isolated in the solid state while the heavier group 10 analogues $[M@Pb_{10}]^{2-}$ (M=Pd, Pt) have been detected in gas-phase experiments[17,22,23]. Even larger clusters such

as $[Pd_2Sn_{18}]^{4-}$[24], based on fused icosahedral units, are also known, but the Pd-Pd separation in this case is too large to allow for meaningful metal-metal bonding. Examples of Zintl clusters supporting directly bonded transition metals are in fact relatively uncommon, although Sevov's $[Ni_3@Ge_{18}]^{4-}$, Dehnen's $[Pd_3@Sn_8Bi_6]^{4-}$ and Fässler's $[Au_3Ge_{18}]^{5-}$ are striking examples[25–27]. In this contribution, we report the synthesis, structure and electronic properties of two members of a family of gold/lead clusters, $[Au_8Pb_{33}]^{6-}$ and $[Au_{12}Pb_{44}]^{8-}$, both isolated as their $[K([2.2.2]crypt)]^+$ salts. Under similar reaction conditions, the corresponding chemistry of Ag leads only to the smaller $[Ag@Pb_{11}]^{3-}$ unit, an observation that naturally raises questions about how the balance between M-M, M-Pb and Pb–Pb bonding controls cluster growth. The wealth of structural data reported here, along with the mass spectrometry of the reaction mixtures and a detailed comparison of the silver and gold chemistry, provides the foundation for a cluster-growth model based on the nido-icosahedral $Au@Pb_{11}$ unit as the fundamental building block.

## Results

**Silver chemistry**. The compound $[K([2.2.2]crypt)]_3[Ag@Pb_{11}]·0.5en$, **1**, was synthesized from the reaction of $K_4Pb_9$ with $(AgMes)_4$ in ethylenediamine (en) solution in the presence of [2.2.2]crypt (full synthetic details are given in the methodology section). Black block-like crystals of **1** contain the anion, $[Ag@Pb_{11}]^{3-}$, along with three $[K([2.2.2]crypt)]^+$ cations (triclinic space group P-1). The anionic component of **1**, $[Ag@Pb_{11}]^{3-}$, shown in Fig. 2, is an approximately $C_{5v}$-symmetric nido-icosahedron, with the Ag center encapsulated by the $Pb_{11}$ cluster (the encapsulation is indicated by the "@" in $[Ag@Pb_{11}]^{3-}$). The Ag–Pb bond lengths to the apical Pb (Pb1 in Fig. 2) and the five Pb atoms of the open face (Pb7-11 in Fig. 2) are all ~3.01 Å while the distances to Pb2-6 are somewhat longer at ~3.09 Å. The Pb–Pb bond lengths vary between 3.15 and 3.30 Å, and are very similar to those reported for the closo clusters $[M@Pb_{10}]^{2-}$ and $[M@Pb_{12}]^{2-/3-}$ discussed in the introduction[16–23]. The characteristic valence electron count of 48 ($4n + 4$) for a nido 11-vertex polyhedron[28] demands a charge of 4– on the $Pb_{11}$ cluster, consistent with the presence of an Ag ion in the +1 oxidation state (i.e., $d^{10}$). Whilst **1** is the only crystalline product obtained from this reaction, the ESI mass spectrum of the reaction mixture (Supplementary Fig. 12) shows prominent peaks for $[AgPb_{10}]^-$ and $[AgPb_{12}]^-$ as well as a somewhat less intense one for $[AgPb_{11}]^-$ itself. The $[AgPb_{10}]^-$ and $[AgPb_{12}]^-$ ions are most likely both closo species, isoelectronic to the isolated compounds $[Ni@Pb_{10}]^{2-}$ and $[Pd@Pb_{12}]^{2-}$.

**Gold chemistry**. The analogous gold chemistry was also carried out using $K_4Pb_9$ as a source of Pb, but now using $Au(Mes)PPh_3$ as the source of the precious metal. Subtle differences in reaction conditions lead to two quite distinct crystalline products, $[K([2.2.2]crypt)]_6[Au_8Pb_{33}]·en$ (**2**) and $[K([2.2.2]crypt)]_8[Au_{12}Pb_{44}]$ (**3**). Heating the reagents in en solution in the presence of [2.2.2]

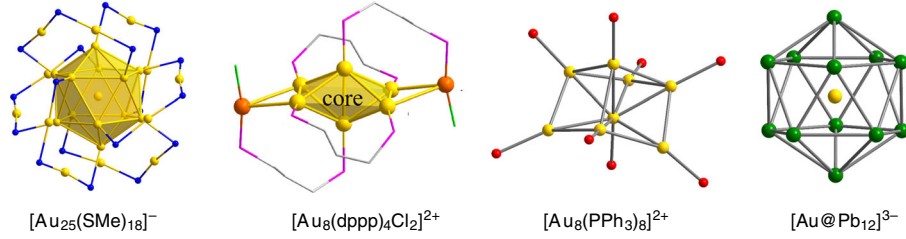

**Fig. 1 Structures of Au-containing clusters.** Structural features of selected gold-containing clusters, $[Au_{25}(SMe)_{18}]^-$, $[Au_8(dppp)_4Cl_2]^{2+}$, $[Au_8(PPh_3)_8]^{2+}$ and $[Au@Pb_{12}]^{3-}$. dppp = bis(diphenylphosphino)propane.

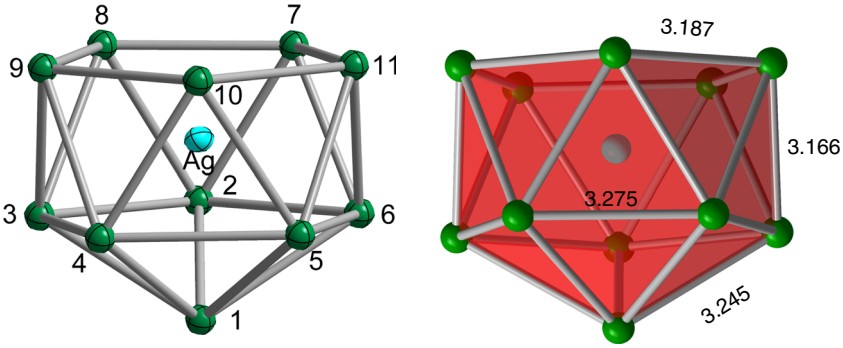

**Fig. 2 Structure of the [Ag@Pb$_{11}$]$^{3-}$ anion in 1.** The bond lengths are the average of all symmetry-related Pb–Pb distances. Thermal ellipsoids are set at 50% probability level.

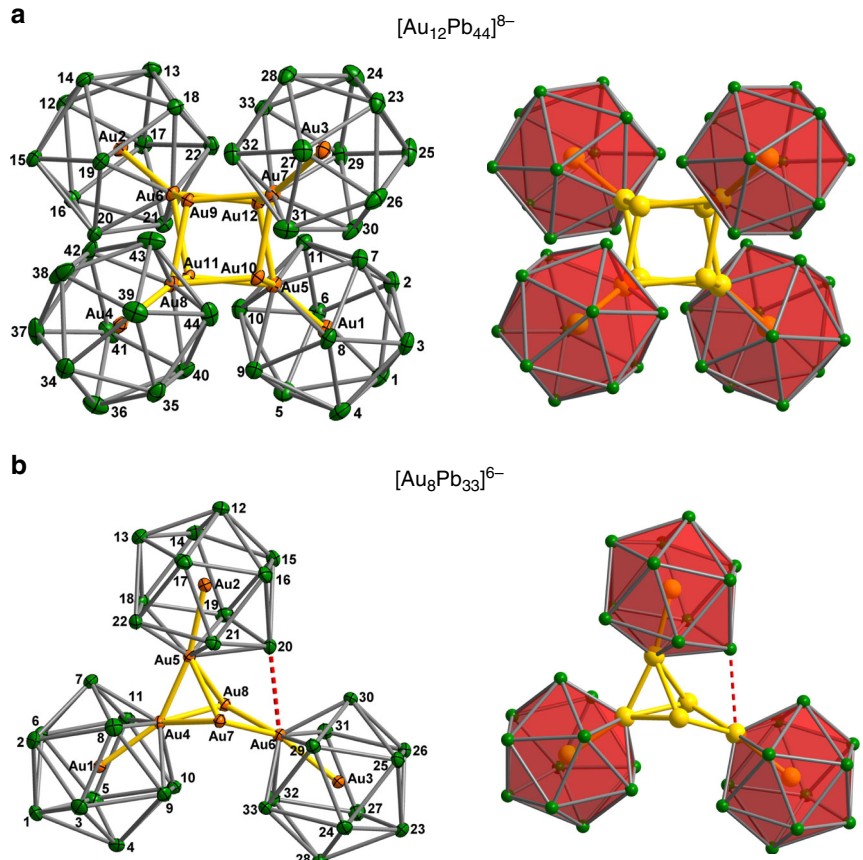

**Fig. 3 Structures of [Au$_{12}$Pb$_{44}$]$^{8-}$ and [Au$_8$Pb$_{33}$]$^{6-}$. a** Structure of the [Au$_{12}$Pb$_{44}$]$^{8-}$ anion in **3**. **b** Structure of the [Au$_8$Pb$_{33}$]$^{6-}$ anion in **2**. Thermal ellipsoids are set at the 50% probability level.

crypt at 60 °C for 3 h led to the formation of the smaller cluster, **2**, in 25% yield (based on Pb content). If, however, the en solvent is removed and the residue re-dissolved in pyridine followed by further heating (4 h at 40 °C), the larger cluster, **3**, is formed in 18% yield. The structures of the two anionic components of **2** and **3**, [Au$_8$Pb$_{33}$]$^{6-}$ and [Au$_{12}$Pb$_{44}$]$^{8-}$, respectively, are shown in Fig. 3.

It is immediately striking that the Au and Pb content of the clusters is segregated, with an inner Au$_x$ core surrounded by an outer Pb$_y$ shell, an observation that is consistent with the low miscibility of the two metals. A closer inspection shows that the two clusters share many common features: both are constructed from Au-centered Pb$_{11}$ *nido*-icosahedra (Au@Pb$_{11}$) similar to those found in **1**, surrounding a core of Au atoms, Au$_5$ and Au$_8$ in

[Au$_8$Pb$_{33}$]$^{6-}$ and [Au$_{12}$Pb$_{44}$]$^{8-}$, respectively. Here and in subsequent discussions, we adopt the nomenclature [m,n] to designate a cluster of composition ([Au@Pb$_{11}$])$_m$(Au)$_n$, where [Au$_8$Pb$_{33}$]$^{6-}$ and [Au$_{12}$Pb$_{44}$]$^{8-}$ constitute the [3, 5] and [4, 8] members, respectively. The four Au@Pb$_{11}$ *nido*-icosahedra in **3** are remarkably similar to each other, and also to the isolated [Ag@Pb$_{11}$]$^{3-}$ cluster: the Pb–Pb bond lengths vary between 3.149 (2) and 3.340(2) Å, slightly longer, on average, than those in typical [M@Pb$_{12}$]$^{n-}$ units (3.10–3.20 Å) and the eleven Au–Pb distances vary between 2.969(2) and 3.174(2) Å. Corresponding values in [Au@Pb$_{12}$]$^{3-}$ lie in the range 3.030(9)–3.093(4) Å[16]. The approximately cubic Au$_8$ core in [Au$_{12}$Pb$_{44}$]$^{8-}$ is shown from two perspectives in Fig. 4a. The Au–Au bond lengths are in the range 2.8941(19)–2.943(2) Å, broadly comparable to those in

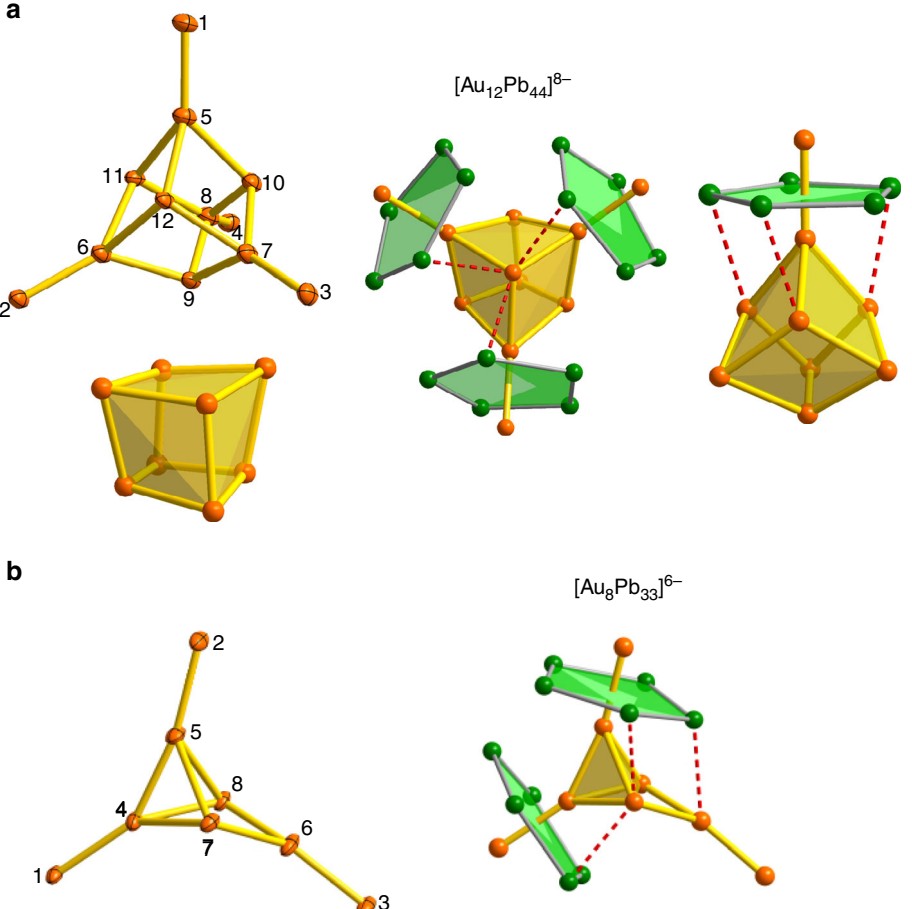

**Fig. 4 Structures of the Au$_x$ cores in [Au$_{12}$Pb$_{44}$]$^{8-}$ and [Au$_8$Pb$_{33}$]$^{6-}$. a** the Au$_{12}$ core in [Au$_{12}$Pb$_{44}$]$^{8-}$ and **b** the Au$_8$ core in [Au$_8$Pb$_{33}$]$^{6-}$. Red dashed lines emphasize the secondary Au...Pb π interactions at ~ 3.60 Å between the capping Au atoms and the Pb$_5$ rings.

metallic gold (2.88 Å)[29], and also in clusters such as [Au$_3$Ge$_{18}$]$^{5-}$, [Sb$_3$Au$_3$Sb$_3$]$^{3-}$ and [Au$_7$(dppp)$_4$]$^{3+}$ [27,30,31]. The open Pb$_5$ faces of the *nido*-icosahedra bind to four corners of the Au$_8$ core (Au5-8, Au–Pb = 2.959–3.030 Å), leading to short distances of 2.803 (2)–2.821(2) Å between the endohedral Au (Au1-4) and the Au atom that completes the icosahedral surface (Au5-8). We refer to these four Au atoms as the 'surface' Au for this reason. The interactions between the Au$_8$ cube and the *nido*-icosahedra are not, however, restricted to the four surface Au atoms directly bound to the open pentagonal faces of the *nido* icosahedra. There are also numerous secondary contacts between the Pb atoms of the icosahedra and the four Au atoms of the Au$_8$ cube that are not bonded directly to the open faces (Au9-12 in Fig. 4—we refer to these as the 'capping' Au atoms). These secondary interactions are shown as dashed red lines in Fig. 4b. The precise bond lengths depend critically on the conformation of the icosahedra, but it is clear that each capping Au atom has secondary contacts at ~3.6 Å with Pb centers on all three neighboring icosahedra and also that each icosahedron has secondary contacts with all three adjacent capping Au atoms (Fig. 4, lower panel). Whilst these secondary interactions are ~0.5 Å longer than the Au–Pb bonds within the icosahedra, they are sufficiently short and sufficiently numerous to play a significant part in maintaining the integrity of the cluster, as we will show in the subsequent analysis of the electronic structure.

The [Au$_8$Pb$_{33}$]$^{6-}$ cluster in **2** is rather less symmetric than **3**, but the icosahedral Au@Pb$_{11}$ units in the two clusters are very similar. Moreover, the structure of the Au$_6$Pb$_{22}$ fragment

(containing Au1,3,4,6,7 and 8 in Figs. 3 and 4) resembles very closely one half of the [Au$_{12}$Pb$_{44}$]$^{8-}$ cluster found in **3**: the Au–Au distances are in the region of 2.9 Å and the secondary interactions between the Pb atoms and the two bridging Au atoms are again apparent. This structural relationship suggests that [Au$_8$Pb$_{33}$]$^{6-}$ can be formulated as a "dimer + monomer" wherein an ([Au@Pb$_{11}$]$_2$)(Au$_4$) unit (the [2, 4] member of the [m,n] series) coalesces with an additional icosahedral Au@AuPb$_{11}$ cluster (the [1, 1] member) to form the [3, 5] cluster. The Au@AuPb$_{11}$ icosahedron bridges the Au$_4$ unit of the [2, 4] fragment in an η$^2$ fashion, *via* both the surface Au atom, Au5, and one of the adjacent Pb atoms, Pb20. The Au4-Au5 bond is, at 2.7609(12) Å, the shortest Au–Au bond in either **2** or **3**, while the Pb20-Au6 distance of 3.4098(12) Å is shorter than any of the other secondary interactions. As a result of this strong secondary interaction, Pb20 is pulled out of the icosahedral surface, leading to four unusually long Pb–Pb bonds between 3.30 and 3.40 Å.

The ESI mass spectrum of the solution from which **3** is isolated (Supplementary Fig. 9) offers some support for the proposal that an Au$_2$Pb$_{11}$ cluster is a common intermediate in the coalescence of the larger clusters. The parent ions of **2** and **3** lie outside the accessible window for ESI mass spectrometry, but the low-mass region (below $m/z$ = 3200) shows prominent peaks due to a number of smaller fragments, the most intense of which are [AuPb$_{10}$]$^-$, [AuPb$_{11}$]$^-$ and [AuPb$_{12}$]$^-$. Although none of these has been crystallized, we suggest that [AuPb$_{10}$]$^-$ and [AuPb$_{12}$]$^-$ are most likely *closo* clusters, isostructural with [Ni@Pb$_{10}$]$^{2-}$ and [Ni@Pb$_{12}$]$^{2-}$, respectively, while [AuPb$_{11}$]$^-$ is probably

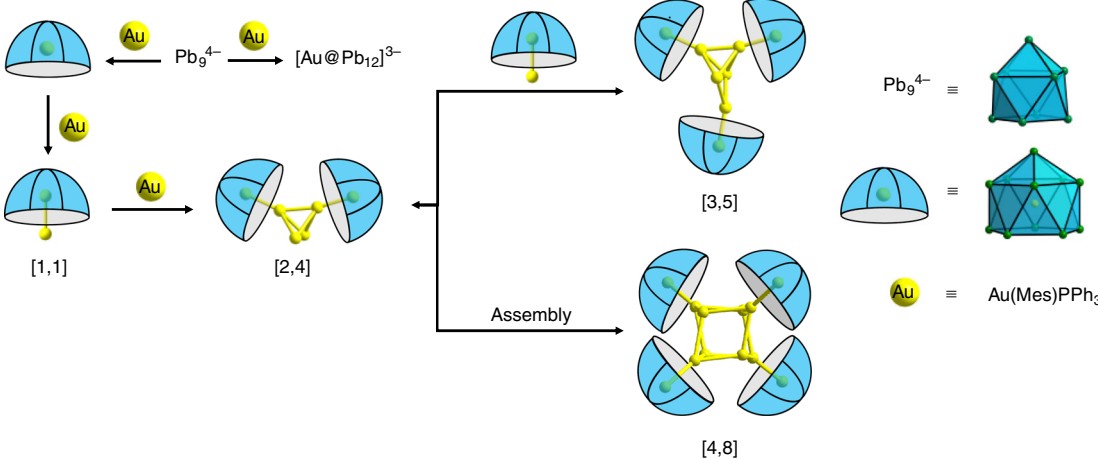

**Fig. 5 Possible pathways leading to cluster growth.** Coalescence of smaller component clusters leads to the assembly of $[Au_8Pb_{33}]^{6-}$ ([3, 5]) and $[Au_{12}Pb_{44}]^{8-}$ ([4, 8]). The [1, 1], [3, 5] and [4, 8] clusters have been observed, either by X-ray crystallography or ESI-MS.

isostructural with **1**. There is, however, an additional small peak at $m/z = 3088.88$ assigned to $[K([2.2.2]crypt)]^+[Au_2Pb_{11}]^{2-}$, precisely the composition of the 'monomeric' [1, 1] fragment in the structure of **2**, suggesting that binding of an additional $Au^+$ ion to the open face of the *nido* $[AuPb_{11}]^{3-}$ icosahedron is possible. It is significant that there is no corresponding peak for $[K([2.2.2]crypt)]^+[Ag_2Pb_{11}]^{2-}$ in Supplementary Fig. 12 (it should be found at $m/z = 2910.68$), indicating that binding of an $Ag^+$ ion to the open face of the $[Ag@Pb_{11}]^{3-}$ is unfavorable, offering a possible explanation for the absence of Ag analogues of the larger clusters **2** and **3**.

Based on a combination of the structural and ESI mass spectroscopic data, we can speculate on possible mechanistic pathways that control cluster growth (Fig. 5). The *nido*-icosahedron $[M@Pb_{11}]^{3-}$ has been structurally characterized for M = Ag and a cluster with the same composition has been observed as a prominent peak in the mass spectrum for M = Au. It seems reasonable, therefore, to propose $[Au@Pb_{11}]^{3-}$ as a likely intermediate in the growth of the larger clusters. The $[Au_2Pb_{11}]^{2-}$ anion observed in the mass spectrum (in combination with a $[K([2.2.2]crypt)]^+$ cation) can then be formed by trapping an $Au^+$ cation at the open face of the *nido* $[Au@Pb_{11}]^{3-}$ cluster to complete the $Au_2Pb_{11}$ icosahedron that is the basic [1, 1] structural unit of both **2** and **3**. In the presence of excess gold (presumably formed by reduction of $Au^+$ with $[Pb_9]^{4-}$), coalescence of two such icosahedra with an $Au_2$ fragment generates the $([Au@Pb_{11}])_2(Au_4)$ unit ([2, 4]) that is common to both **2** and **3**. This fragment may then either dimerize to form **3** ([4, 8]) or react with a third $[Au@AuPb_{11}]^{2-}$ fragment to form **2** ([3, 5]). In support of this hypothesis, we have observed that heating an isolated sample of **2** for 3 h at 60 °C leads to the formation of **3**, presumably via de-coordination of the bridging icosahedron followed by coalescence of two dimer units (Supplementary Fig. 8). A plausible alternative cluster growth pathway might involve the bonding of multiple copies of the fundamental ligand unit, $[Au@Pb_{11}]^{3-}$, to pre-formed $[Au_5]^{3+}$ and $[Au_8]^{4+}$ clusters to generate **2** and **3**, respectively. We have, however, found no evidence to support the formation of such large naked gold clusters under the prevailing reaction conditions, so we favor the simpler scheme below where the $Au@AuPb_{11}$ unit, for which there is experimental evidence, is the common intermediate. In either case, the identity of the dominant isolated product will necessarily be very sensitive to the concentrations of free Au, and so, inevitably, to subtle variations in temperature and solvent polarity.

**Electronic structure analysis.** In order to gain further insight into the bonding in the anionic clusters in **1**, **2** and **3**, and the possible pathways that lead to their formation, we have turned to density functional theory. In Fig. 5 we proposed the $[Au@Pb_{11}]^{3-}$ unit as the initial product of the reaction between $Au(Mes)PPh_3$ and $K_4Pb_9$, based on the twin observations that (a) the corresponding mono-anion is present in the mass spectrum and (b) the Ag analogue can be isolated in the solid state. This makes the isolated $[M@Pb_{11}]^{3-}$ units a logical place to start our analysis of the electronic structure. The optimized structures of $[Ag@Pb_{11}]^{3-}$ and $[Au@Pb_{11}]^{3-}$ shown in Fig. 6 are rigorously $C_{5v}$ symmetric, with the group 11 metal endohedrally encapsulated in both cases. Alternative structures where the group 11 metal occupies a site on the surface of the cluster, completing an empty $[MPb_{11}]^{3-}$ icosahedron, prove to be marginally less stable for both Ag and Au (see *exo*-$[MPb_{11}]^{3-}$ in Supplementary Table 6). For $[Ag@Pb_{11}]^{3-}$, where crystallographic data are available as a benchmark, the Ag–Pb and Pb–Pb bonds are overestimated by ~0.1 Å: this is a common observation in highly anionic clusters of this type, and probably reflects the fact that the confining influence of the cationic lattice is modeled only by a high-dielectric continuum in the computational experiment. The important frontier orbitals of $[Au@Pb_{11}]^{3-}$, also shown in Fig. 6, are dominated by Pb $5p$ character on the open face of the *nido* cluster. The HOMO−4 is totally symmetric ($a_1$) while the HOMO/HOMO−1 is a degenerate pair ($e_1$). The fragment is isolobal to the $[C_5H_5]^-$ ligand and also to *nido*-$[B_{11}H_{11}]^{4-}$, the coordination chemistry of which is well established through complexes such as $[(\eta^5\text{-}B_{11}H_{11})_2Ni]^{4-}$ [32], and has the capacity to act as a 6-electron donor to the $Au_x$ core. The optimized structure of $[Au@AuPb_{11}]^{2-}$ is also $C_{5v}$-symmetric with an Au–Au distance of 2.75 Å, very similar to those in all the icosahedral units in $[Au_8Pb_{33}]^{6-}$ and $[Au_{12}Pb_{44}]^{8-}$. The bond orders between the surface $Au^+$ ion and the Pb atoms of the open face are 0.25, compared to values between 0.08 and 0.24 for Au–Pb bonds to the endohedrally encapsulated $Au^+$ ion. By contrast, the Au–Au bond order is only 0.05, suggesting that direct Au–Au bonding within the icosahedron is relatively weak despite the short Au–Au distance: this is consistent with a closed-shell $d^{10}$ configuration at both metals. The Kohn–Sham interaction diagram in Fig. 6 summarizes the key features of the interaction of the $Au^+$ cation with the *nido* $[Au@Pb_{11}]^{3-}$ fragment. The bonding is dominated by orbitals of local σ symmetry ($a_1$ in $C_{5v}$), i.e., the donation of charge from the HOMO−4 of $[Au@Pb_{11}]^{3-}$ to the $6s$ orbital of $Au^+$. The doubly degenerate HOMO/HOMO−1 ($e_1$) has π

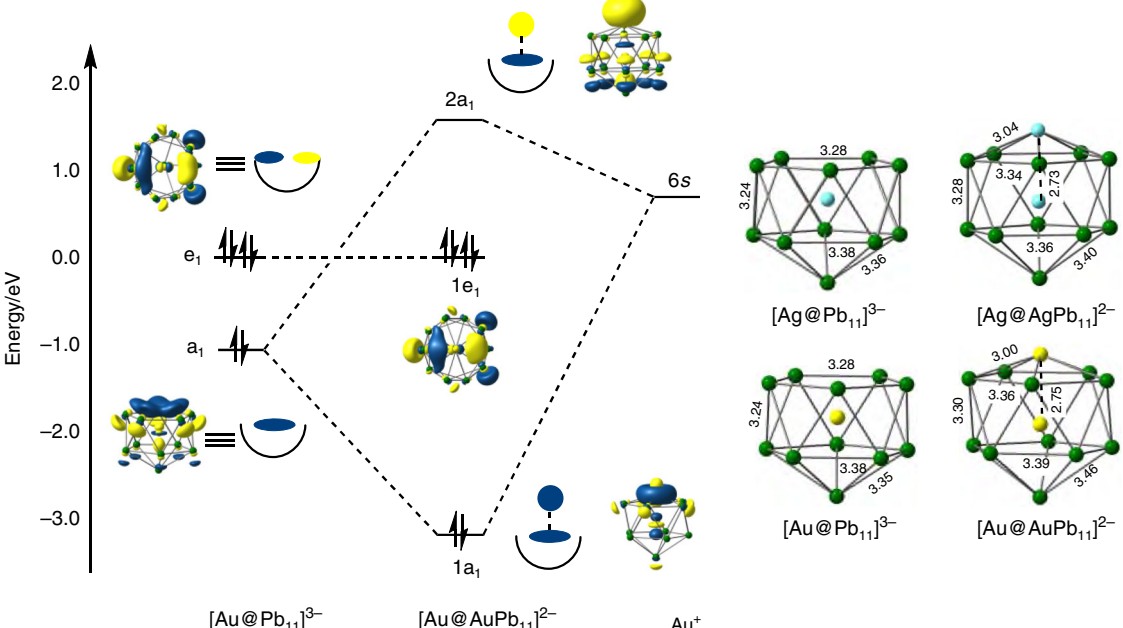

**Fig. 6 Electronic structure analysis of [M@Pb₁₁]³⁻ and [M@MPb₁₁]²⁻.** Optimized structures of $[M@Pb_{11}]^{3-}$ and $[M@MPb_{11}]^{2-}$, and a Kohn–Sham molecular orbital diagram showing the interaction between $Au^+$ and the *nido*-icosahedral fragment $[Au@Pb_{11}]^{3-}$. The doubly degenerate $e_1$ orbitals are viewed down the principal axis.

symmetry with respect to the fivefold rotational axis, but it plays no part in the bonding due to the absence of vacant orbitals of appropriate symmetry on $Au^+$. The $1e_1$ HOMO therefore remains non-bonding and high in energy even after binding of the surface $Au^+$ ion: we will show later that the continuing availability of this doubly degenerate orbital is vital to the formation of the secondary Au–Pb bonds. The typical valence electron count for a *closo* icosahedron is 50, made up of 26 skeletal electrons and 24 radially directed lone pair electrons, 2 on each vertex. The count for $[Au@AuPb_{11}]^{2-}$, in contrast, is only 48, and it is clear from Fig. 6 that the 'missing' pair of electrons is taken from a radially directed hybrid with dominant Au 6s character (the $2a_1$ LUMO) rather than a skeletal bonding orbital, and so the *closo* count of 26 remains intact. In this sense $[Au@AuPb_{11}]^{2-}$ can be viewed either as a *nido* $[Au@Pb_{11}]^{3-}$ cluster capped by an $Au^+$ cation or as a *closo* $Au@AuPb_{11}$ icosahedron with a missing lone pair—both perspectives are consistent with an electron count of 48. We can quantify the energetic significance of the interactions within $a_1$ symmetry by performing an energy decomposition analysis based on the fragmentation of $[Au@AuPb_{11}]^{2-}$ into $[Au@Pb_{11}]^{3-}$ and $Au^+$ (see Supplementary Table 7 for details). This indicates an energetic contribution of –4.58 eV for the interaction in $[Au@AuPb_{11}]^{2-}$ compared to only –2.87 eV in the Ag analogue, $[Ag@AgPb_{11}]^{2-}$. The difference of almost 2 eV reflects the strong relativistic stabilization of the 6s orbital in Au, and the weak binding of $Ag^+$ to the open face may, ultimately, be the underlying cause of the absence of larger clusters in the Ag chemistry.

The important features of the bonding in the larger clusters are most easily approached through an analysis of $[Au_{12}Pb_{44}]^{8-}$, where the relatively high symmetry simplifies the interpretation of the electronic structure. The cluster in the crystal is approximately $D_{2d}$ symmetric, but small deviations inevitably arise due to the low symmetry of the crystalline environment. We have, therefore, adjusted the coordinates to impose strict $D_{2d}$ symmetry in a structure that is closest, in a least-squares sense, to the geometry of the cluster in the crystal (see Supplementary Fig. 16 for details). Given the overestimation of Pb–Pb bond

lengths that we encountered for the tri-anionic $[Ag@Pb_{11}]^{3-}$ cluster, we have made no attempt to further optimize the structure of $[Au_{12}Pb_{44}]^{8-}$. Single point calculations on the $D_{2d}$-symmetrized structure indicate the presence of a near degenerate triplet of orbitals, $e + b_2$, in the frontier region, over which two electrons are distributed (Fig. 7). The near degeneracy arises because the $Au_8$ core is not strongly distorted from a perfect tetrahedron, in which limit the $e + b_2$ manifold correlates with triply degenerate $t_2$. By distributing two electrons over these three orbitals we can converge on two triplet states, $^3A_2$ and $^3E$, ($e^2$ and $e^1b^1$ configurations, respectively) and a closed-shell singlet ($^1A_1$, $b_2{}^2$), all of which lie within 0.03 eV. Given the well-documented limitations of DFT in identifying ground-state multiplicities[33,34], these energies are too close to allow for a definitive conclusion on the ground spin state of $[Au_{12}Pb_{44}]^{8-}$. The following analysis is based on the singlet, although very similar features emerge in the other two low-lying states. The bond orders for the Au–Au and Au–Pb bonds within the individual icosahedra are very similar to those in the isolated $[Au@AuPb_{11}]^{2-}$ fragment (~0.06 and ~0.13–0.16, respectively). Perhaps more surprisingly, bond orders for the Au–Au bonds within the $Au_8$ cube are also small (0.09–0.13), and in fact are of similar magnitude to the secondary Au–Pb interactions between the icosahedra and the capping Au atoms alluded to previously (0.08–0.11). Given the large number of these secondary interactions (on average, 3 per capping Au atom, 3 per icosahedron, Fig. 4), it seems likely that they are more influential in determining the structure than direct Au–Au bonding. This might account for the very different structure adopted by the $[Au_8]^{4+}$ core in **2** compared to the bicapped octahedral in $[Au_8(dppp)_4X_2]^{2+}$[15], where Au–Au bonding is clearly the dominant structural influence.

A schematic molecular orbital diagram where the cluster is decomposed into four *nido* $[Au@Pb_{11}]^{3-}$ ligands and an approximately cubic $[Au_8]^{4+}$ core is presented in Fig. 7. The calculations were performed in $D_{2d}$ symmetry, and a full analysis is presented in the Supplementary Figs. 16 and 17 and Supplementary Table 8, but for the sake of clarity we find it convenient here to adopt the symmetry labels of the higher $T_d$

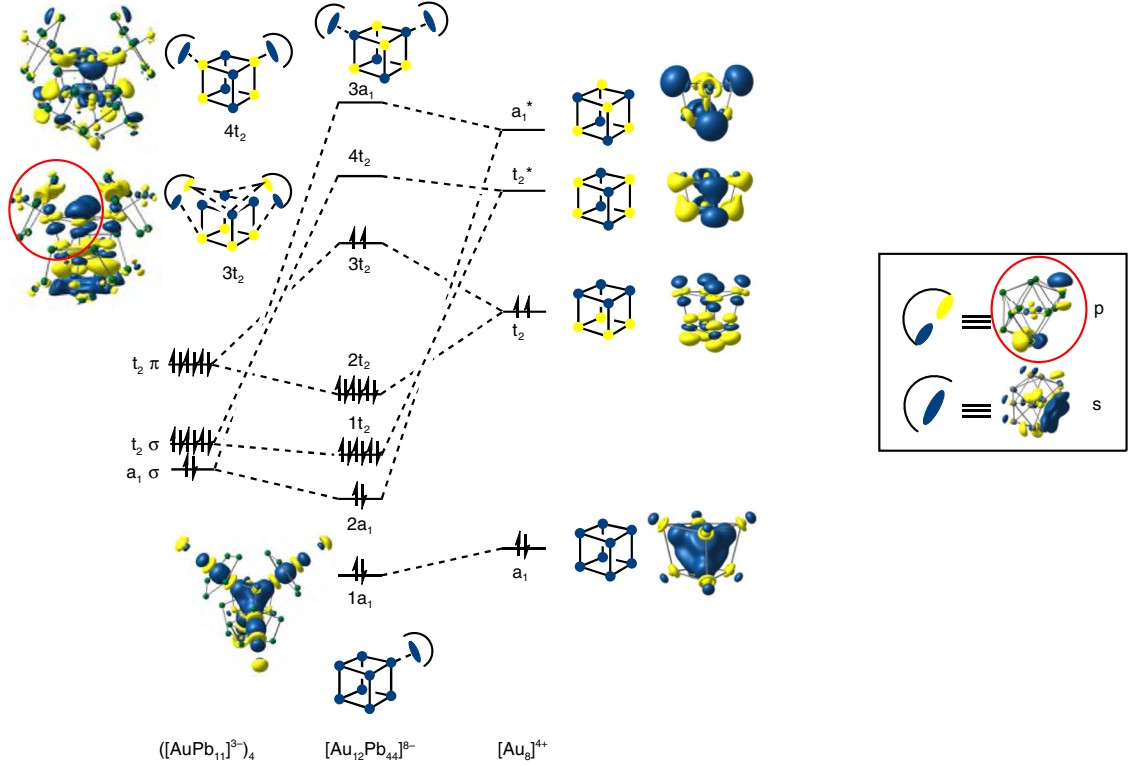

**Fig. 7 Electronic structure of [Au₁₂Pb₄₄]⁸⁻.** Schematic MO diagrams for $[Au_{12}Pb_{44}]^{8-}$, with orbitals labeled according to $T_d$ point symmetry. The majority of the Pb atoms have been removed from the iso-surface plots for clarity. Cartoon depictions of the orbitals are provided to guide the eye.

point group. The frontier region for the $[Au_8]^{4+}$ fragment can be understood in terms of the interactions of 8 $s/d_{z^2}$ hybrids, which generate bonding and antibonding linear combinations, $a_1/a_1^*$ and $t_2/t_2^*$. The bonding $a_1$ orbital is strongly stabilized and doubly occupied, and this pair of electrons is primarily responsible for the integrity of the $Au_8$ unit in $[Au_8]^{4+}$ and also in **3** itself. The remaining two valence electrons then occupy the $t_2$ orbital, driving a first-order Jahn–Teller instability which accounts for the slight distortion of the $Au_8$ core in $[Au_{12}Pb_{44}]^{8-}$ from perfect tetrahedral symmetry. This distortion is, however, a minor feature that does not negate the value of presenting the analysis in the higher point group. When the four *nido* $[Au@Pb_{11}]^{3-}$ ligands are introduced, we can identify two distinct types of interactions with the $[Au_8]^{4+}$ cluster (see inset in Fig. 7). First, the HOMO–4 of the ligands ($a_1$ in Fig. 6) generate linear combinations ($a_1 + t_2$) with local σ symmetry that can overlap directly with the corresponding linear combinations of $s/d_{z^2}$ hybrids on the Au atoms bonded directly to the open $Pb_5$ faces, in precisely the same way as the $s/d_{z^2}$ hybrid on the single Au atom did in $[Au_2@Pb_{11}]^{2-}$ (see the cartoon representations of the $3a_1$ and $4t_2$ orbitals in Fig. 7). The second mechanism involves the doubly degenerate HOMO/HOMO–1 of the ligand which has local π symmetry and generates linear combinations of $e + t_1 + t_2$ symmetry, the last of which can overlap with a $t_2$-symmetric linear combinations of $s/d_{z^2}$ hybrids on the three adjacent capping Au atoms. The π character of this pathway is shown in the cartoon representation of the $3t_2$ orbital of $[Au_{12}Pb_{44}]^{8-}$ in Fig. 7, and the contours of the corresponding isosurface on the *nido* $[Au@Pb_{11}]^{3-}$ ligands (highlighted in the red circles in Fig. 7), show the clear fingerprint of the HOMO/HOMO–1. The bonding of the $[Au@Pb_{11}]^{3-}$ ligand to the $Au_8$ core is therefore made up of two quite distinct components—σ bonding to the Au atoms that bind to the center of the $Pb_5$ faces

and secondary π bonding to the three adjacent capping Au atoms. We can estimate the relative importance of the two components by performing an energy decomposition analysis, noting that the σ-type interactions are mediated primarily by the $a_1^*$ and $t_2^*$ orbitals of the $Au_8$ core while the π-type interactions involve primarily the partially occupied $t_2$ orbital. By successively removing virtual fragment orbitals from the basis (using the removefragorbitals option in ADF), we can therefore associate distinct energetic contributions to the σ and π pathways. Eliminating the four virtual orbitals on $[Au_8]^{4+}$ that mediate the σ pathway ($a_1^*$ and $t_2^*$) reduces the total interaction energy by 2.4 eV, while further closing down the secondary π pathway by eliminating the two unoccupied components of $t_2$ leads to an additional loss of 3.2 eV. By this measure, it appears, therefore, that the secondary π-type interactions make a dominant contribution to the overall stability of the cluster: they are particularly strong because they allow for electron transfer from the high-energy HOMO and HOMO–1 of the ligand to the $Au_8$ core.

## Discussion

The common structural elements of the two Au clusters reported here, $[Au_8Pb_{33}]^{6-}$ and $[Au_{12}Pb_{44}]^{8-}$, suggest that both are formed by fusion of icosahedral $[Au@Pb_{11}Au]^{2-}$ fragments which can, in turn, be generated from the precursor ligand $[Au@Pb_{11}]^{3-}$ by trapping of $Au^+$. Although neither of these smaller fragments has been crystallized, the Ag analogue $[Ag@Pb_{11}]^{3-}$ can be isolated from the corresponding reactions with $(AgMes)_4$, while fragments corresponding to both $[Au@Pb_{11}]^-$ and $[Au_2Pb_{11}]^-$ have been observed by mass spectrometry. In contrast we find no evidence for the formation of $[Ag_2Pb_{11}]^-$ and we suspect that the instability of this species is the root cause of the absence of larger silver/lead

clusters. The bonding of the *nido* $[Au@Pb_{11}]^{3-}$ ligands to the $[Au_8]^{4+}$ core of **2** is made up of two distinct components: one Au is bound to the center of the open $Pb_5$ face through a σ-symmetric interaction while orbitals of π symmetry interact with the three adjacent Au atoms. Overall, then, each $[Au@Pb_{11}]^{3-}$ ligand has bonding interactions with four Au atoms of the core. Density functional theory indicates that the secondary interactions are very significant energetically, and in fact contribute more to the integrity of the cluster than direct Au–Au bonding. The importance of these secondary π-type Pb…Au interactions presents an interesting parallel to gold thiolate cluster family, where the 'divide and protect' model has anionic 'staple' ligands containing $[(RS)Au^+(SR)]^-$ and $[(RS)Au^+(SR)Au^+(SR)]^-$ binding to the zerovalent $Au_x$ core via two terminal sulfurs. In the Au/Pb cluster family, the $[Au@Pb_{11}]^{3-}$ ligands play a similar role in that they bind to the $Au_n$ core ($n = 5$ and 8 in $[Au_8Pb_{33}]^{6-}$ and $[Au_{12}Pb_{44}]^{8-}$, respectively) via the open $Pb_5$ face of the *nido* $Pb_{11}$ units. The cluster growth model shown in Fig. 5 allows us to speculate on what other members of the Au/Pb family might be accessible. The next obvious stages in cluster growth would be the [5, 11] cluster, a hexa-capped trigonal bipyramid with overall composite $[Au_{16}Pb_{55}]^{10-}$, and a [6, 14] octa-capped octahedron, $[Au_{20}Pb_{66}]^{12-}$. Whilst the progressive 2−increase in anionic charge at each step will certainly terminate the series before very large gold cores can be reached, these larger clusters may be accessible under reducing conditions where Au is present in excess.

## Methods
**Synthesis of [K([2.2.2]crypt)]₃[Ag@Pb₁₁]·0.5en (1)**. In a 10 mL vial, 150 mg (0.074 mmol) of $K_4Pb_9$ and 100 mg (0.27 mmol) of 4,7,13,16,21,24-hexaoxa-1,10-diazabicyclo[8.8.8]hexacosane (abbreviated henceforth as [2.2.2]crypt) were dissolved in ethylenediamine (2.5 mL). After stirring for 1 h, the black-green solution was filtered onto 40 mg (0.04 mmol) of (AgMes)₄. After 3 h at room temperature, the resulting deep-black solution was filtered through glass wool and transferred to a test tube. After 15 days, black block-like crystals of [K([2.2.2]crypt)]₃[Ag@Pb₁₁]·0.5en was obtained by layering with toluene (3 mL) (25% crystalline yields based on Pb). ESI-MS of the products of the reaction with (AgMes)₄ are shown in Supplementary Figs. 12–14. The energy dispersive X-ray (EDX) spectrum of **1** is shown in Supplementary Fig. 15. Note: all Ag-related reactions should be protected from light to avoid decompositions.

**Synthesis of [K([2.2.2]crypt)]₆[Au₈Pb₃₃]·en (2)**. In a 10 mL vial, 150 mg (0.074 mmol) of $K_4Pb_9$ and 100 mg (0.27 mmol) of [2.2.2]crypt were dissolved in ethylenediamine (2.5 mL). In a second vial, 150 mg (0.25 mmol) Au(Mes)PPh₃ was dissolved in 0.5 mL toluene. The toluene solution was added to ethylenediamine solution dropwise while stirring vigorously. After 3 h at 60 °C, the resulting red-black solution was filtered through glass wool and transferred to a test tube. After 7 days, black plate-like crystals of [K([2.2.2]crypt)]₆[Au₈Pb₃₃]·en were obtained together with black rod-like crystals of [K([2.2.2]crypt)]₄[Au₄Pb₂₂] by layering with toluene (3 mL) (25% crystalline total yields based on Pb). ESI-MS of the products of the reaction with Au(Mes)PPh₃ are shown in Supplementary Figs. 9–11. EDX spectra of **2** and **3** are shown in Supplementary Fig. 15.

**Synthesis of [K([2.2.2]crypt)]₈[Au₁₂Pb₄₄] (3)**. In a 10 mL vial, 150 mg (0.074 mmol) of $K_4Pb_9$ and 100 mg (0.27 mmol) of [2.2.2]crypt were dissolved in ethylenediamine (2.5 mL). In a second vial, 150 mg (0.25 mmol) Au(Mes)PPh₃ was dissolved in 0.5 mL toluene. The toluene solution was added to ethylenediamine solution dropwise and stirred 3 h at 60 °C while stirring vigorously. After removal of the ethylenediamine solvent, the residue was re-dissolved in pyridine (3 mL) and, after heating for 4 h at 40 °C, the resulting dark purple solution was filtered through glass wool and transferred to a test tube. After 2 weeks, black block-like crystals of [K([2.2.2]crypt)]₈[Au₁₂Pb₄₄] was obtained by layering with toluene (3 mL) (18% crystalline yield based on Pb). In a separate experiment, a crystalline sample of **3** was also obtained starting from an isolated sample (20 mg) of **2** dissolved in pyridine (1.0 mL) in an NMR tube. The resulting red-brown solution was heated at 60 °C for 3 h and then layered by toluene (1.0 mL) to allow for crystallization. Black block-like crystals of [K([2.2.2]crypt)]₈[Au₁₂Pb₄₄] were isolated after two weeks (25% yield based on **2**).

**Single crystal X-ray diffraction data analyses**. The available data for **1** and **2** were refined successfully against the structural models, as measured by $R_1$ and $wR_2$ values of less than 0.07 and 0.18, respectively. In contrast, all attempts to obtain the high-quality X-ray diffraction data for compound **3** were unsuccessful due to the

absorption of the Cu light source ($\lambda = 1.54184$ Å) by the Au and Pb elements of the cluster and its large unit cell volume. Despite this, one data set of reasonable quality was obtained but still contains a relatively large final R value of 15.06%. All diffraction methods were carried out at 100 K. A summary of the crystallographic data for these complexes is listed in Supplementary Table 1, and selected bond distances are given in Supplementary Tables 2–4 for compound **1**, **2**, and **3**, respectively. Photographs of the crystals are shown in Supplementary Fig. 1, while unit cells and asymmetric units for **1**, **2** and **3** are shown in Supplementary Figs. 2–7.

**Computational details**. All density functional calculations were performed using the Amsterdam Density Functional (ADF) software package, version 2017.11[35–37]. The Perdew Burke Ernzerhof (PBE) functional[38] was used in conjunction with a polarized triple-zeta (TZP) basis on Ag, Au, and Pb[39]. Core orbitals up to and including 3d (Ag) and 4d (Au, Pb) were treated as core for both atoms ("small" core option in ADF). Scalar relativistic corrections were included using the Zero-Order relativistic approximation (ZORA)[40–42]. The confining effects of the cation environment was mimicked using a continuum solvent model with dielectric constant of 78.39[43]. Where geometries were optimized, the gradient algorithm of Versluis and Ziegler was employed[44]. The energy decomposition was performed according to the scheme proposed by Ziegler and Rauk (Supplementary Tables 7 and 8)[45]. Bond orders are computed according to the scheme of Nalewajski and Mrozek (Supplementary Table 9)[46]. Full details of the optimized geometries and the geometries used in the single point calculations on $[Au_{12}Pb_{44}]^{8-}$ are summarized in Supplementary Table 6 and Supplementary Fig. 16. The effects of permuting Au and Pb positions in $[Au_{12}Pb_{44}]^{8-}$ are collected in Supplementary Fig. 17.

## Data availability
The X-ray crystallographic of compounds **1**, **2**, and **3** reported in this study have been deposited at the Cambridge Crystallographic Data Centre (CCDC) under deposition numbers 1972423–1972425. These data can be obtained free of charge from the Cambridge Crystallographic Data Centre via www.ccdc.cam.ac.uk/data_request/cif. The authors declare that all other data supporting the findings of this study are available within the paper (and its Supplementary Information files).

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

## Acknowledgements

The relevant work was supported by the National Natural Science Foundation of China (21971118 and 21722106 to Z.-M.S.). H.W.T.M. thanks the EPSRC for support through the Centre for Doctoral Training, Theory and Modeling in Chemical Sciences under Grant EP/L015722/1.

## Author contributions

Z.-M.S. and J.E.M conceived the project. Z.-M.S. designed the experiments. C.-C.S. conducted the synthesis and C.-C.S. and L.Q. performed the crystallography. J.E.M and H.W.T.M. performed the quantum chemical calculations and analyzed the data. All authors co-wrote the manuscript.

## Competing interests
The authors declare no competing interests.
