## [Peer Review File · Nature Communications]

REVIEWER COMMENTS

Reviewer #1 (Remarks to the Author):

The compounds reported in this manuscripts I find spectacular and fully deserving to be published in this journal. The results are very novel including the Au- and Ag-centered eleven-atom nido species in the Pb-group, the gold cores, and the overall aggregation of such species. The synthesis and characterization are very well carried out and their rationalization based on electronic structure and bonding is well explained. Having said this, I have one major criticism and one major technical issue.

1) In the introduction, the authors discussed the new species almost exclusively in the context of the known and popular nowadays gold-thiolate clusters. This maybe OK as a sales pitch, but the reported clusters are clearly far from having anything in common with these know species, they are not in their family. At the same time, there is not much discussion about the relation to the other Zintl ions in the Pb-group and their stabilization by transition metals, both endohedrally and as part of the cluster surface, e.g. various naked and substituted Sn and Ge clusters stabilized with Pd, Ni, Pt. The new species clearly belong to this family and should be presented also in that context. I suggest the authors reduce significantly the gold-thiolate cluster talk and add more discussion on the related Zintl ions (not just Pb-based!).

2) The technical issue concerns $[\text{Au}@\text{AuPb}_{11}](2-)$. For simplicity we can assume gold to be a main-group element with one electron (nothing changes if we work with two gold atoms each with 11 electrons). It is well established that an icosahedron, a closo-cluster, requires $4n+2$ valence electrons where $n = 12$, and this comes to $4 \times 12 + 2 = 50$. However, the number of electrons available from $[\text{Au}@\text{AuPb}_{11}](2-)$ is 2×1 (from Au) + 11×4 (from Pb) + 2 (from charge) = 48, or two electrons short. The reason for this deviation needs to be explained in detail in the manuscript with specific focus on the orbital that becomes empty in $[\text{Au}@\text{AuPb}_{11}](2-)$ and the specific reason for that compared to normal icosahedron. It should be pointed out that, at the same time, the nido-species $\text{Ag}@\text{Pb}_{11}$ has the correct charge of 3- following the standard electron counting rules.

Reviewer #2 (Remarks to the Author):

This manuscript describes the synthesis and structures of exciting new Ag-Pb and Au-Pb clusters that demonstrate new bonding motifs and structure types. The compounds are characterized through X-ray, DFT and some MS experiments. These large intermetalloid clusters are on the interface of molecular and nanoscience. I recommend publication in Nature Commun. after minor revisions.

My only major concern is that there is a lot of speculation about structures from partial crystal structures and mass spec fragments. I think much of this speculation, and the partial crystal structure, could be removed and the paper would be much more cohesive.

Some specific questions:

1. How do the authors know that Au and Pb segregate? Chemically it makes chemical sense but it is virtually impossible to distinguish Pb from Au by X-ray.
2. Ag@Pb₁₁ is a nido cluster. The Au@Pb₁₁Au icosahedra are not nido. They are 24 electron hypercloso by my count. There might be a nido Au@Pb₁₁ cluster involved in the synthesis but the skeletal electron count of the observed fragments should be specified.
3. I'm not sure I see how the Au@Pb₁₁ ligands would be "tripodal". They are bound eta-5 to the Au atoms of the gold cores. This should be clarified.
4. Au₁₂Pb₄₄ should be subject to a Jahn Teller distortion. Do the authors have any thoughts on this?
5. References: There have been some exciting new Au clusters out of Germany (Tubingen) that might be included in the discussion since they are quite relevant to the compounds at hand. Also, the series of M@Pb₁₂ clusters (Co, Rh, Ir) has recently appeared in Chem. Eur. J. and should be included.

Referee 1

1. We acknowledge the point that the clusters do not resemble closely the thiolate-based species. Our intention here (as with our other papers) is not to treat compounds in isolation, but rather to establish connections between them and what is known in the literature, and we believe that there is some synergy between the secondary Au-Pb interactions here and the divide and protect motifs in the thiolates. Nevertheless, we accept that this point was over-emphasised in the introductory comments, and we have reduced its importance in the revised version. We have also expanded the discussion to place the new clusters more firmly in the Zintl domain.
2. With regard to the comment on $[\text{Au}_2\text{Pb}_{11}]^{2-}$, Pb is isoelectronic with Au^{3-} , so the referee is absolutely correct to note that the total count is two fewer than the *closo* value of 50. The 'missing' 2 electrons are what would be an outwardly directed 'lone pair' of electrons in 50-electron $[\text{Au}@\text{Pb}_{12}]^-$, or indeed $[\text{Au}_2\text{Pb}_{11}]^{4-}$ (the vacant orbital $2a_1$ in Figure 7). Thus $[\text{Au}_2\text{Pb}_{11}]^{2-}$ is not a complete *closo* icosahedron – we prefer to think of it as a *nido* $[\text{AuPb}_{11}]^{3-}$ cluster capped by Au^+ - this formulation makes the absence of a lone pair on Au more obvious. We have added a note to make this point clear (and see also response to referee 2's third point, below).

Referee 2

1. We acknowledge the point regarding the discussion of growth mechanisms. This is necessarily speculative but also one of the key discussion points in contemporary cluster chemistry, so we believe that it is important to at least address the question here, given that we have isolated clusters with different nuclearity. Nevertheless we accept that the partial crystal structure of $[\text{Au}_4\text{Pb}_{22}]^{4-}$ distracts from the focus of the manuscript, so we have removed that element and downgraded the discussion of growth pathways.
2. We agree that it is very difficult to differentiate Au and Pb from a purely crystallographic perspective. However, the coordination environments of the Pb and Au centers in the structure are very different: all 44 sites assigned as Pb have pentagonal pyramidal environments, very typical of atoms such as Pb that bear a lone pair of electrons. Conversely, the sites assigned as Au are either 12-coordinate (endohehdral), 8 coordinate (the surface Au atoms) or 7-coordinate (the capping Au atoms, where we include the secondary interactions in this count). There is no obvious place to accommodate a lone pair at any of the sites assigned to Au. Moreover, the $\text{M}@\text{Pb}_{11}$ motif is well established here and elsewhere, and it seems reasonable to propose that the $\text{A}@\text{B}_{11}$ units in the crystal structures of the larger clusters are also $\text{M}@\text{Pb}_{11}$ – the Pb-Pb bond lengths are almost identical to this on $[\text{Ag}@\text{Pb}_{11}]^{3-}$, where the distinction between Ag and Pb is clear. The distances between the atoms proposed to be Au are also substantially shorter than between those we propose to be Pb. For instance, the average Pb-Pb bond length in $\text{M}@\text{Pb}_{12}$ series is around 3.23 Å, which is comparable with those in the Au-Pb clusters (3.248 Å in $[\text{Au}_8\text{Pb}_{33}]^{6-}$, 3.247 Å in $[\text{Au}_{12}\text{Pb}_{44}]^{8-}$). The average Au-Pb distances (3.049 Å in $[\text{Au}_8\text{Pb}_{33}]^{6-}$, 3.069 Å in $[\text{Au}_{12}\text{Pb}_{44}]^{8-}$) and Au-Au distances (2.856 Å in $[\text{Au}_8\text{Pb}_{33}]^{6-}$, 2.893 Å in $[\text{Au}_{12}\text{Pb}_{44}]^{8-}$) are notably shorter than the Pb-Pb bonds.

In light of the reviewer's comments, we have conducted a further series of DFT calculations where we have taken the proposed structure and permuted sets of Au and Pb atoms. In order to conserve the D_{2d} symmetry, the permutations have been done 4 at a time: for example we have permuted all 4 surface Au atoms with 4 Pb atoms from different positions in the four Pb_{11} units. The results are now included in the SI, but, in short, the structure as proposed is by some distance the most stable. Placing a Pb atom at either the endohedral or surface sites is energetically very costly (~2.0 - 2.5 eV per permutation), reflecting the fact that coordination numbers at these sites are high (and approximately spherical) so there is no place to accommodate the Pb lone pair. It is somewhat less costly to place a Pb atom at the capping sites, presumably because the long secondary interactions on one side of the atom offer some possibility to accommodate the lone pair. Nevertheless, this structure is still 0.25 eV less stable per Au/Pb permutation).

3. With regard to the electron counting in $[\text{Au}@Pb_{11}\text{Au}]^{2-}$, we have addressed this issue in part in our response to point 2 of the first reviewer. We agree that the total electron count of 48 would, in principle, be consistent with a *hypo-closo* count. However, the missing pair of electrons from the *closo* count come from a radially directed orbital on the surface Au (see the empty $2a_1$ orbital in Figure 7). In Corbett's work on Ti_x clusters, the '*hypo-closo*' count is associated with cases where a *trans*-cluster bond forms – in other words the vacant orbital has E-E antibonding character. This is not the case here, so we do not believe that the *hypo-closo* label is appropriate. Rather we believe it is better to think of it as a 26-skeletal-electron *nido* $[\text{AuPb}_{11}]^{3-}$ cluster, capped by a cation which delivers no additional electrons to the cluster (a 'fat proton'). We have augmented the discussion to reflect these points, given that our perspective was not obvious to either of the referees.
4. We agree that the 'tripodal' terminology was not correct – we were alluding to the presence of 3 pairs of electrons on the open Pb_5 faces, but the reviewer's suggestion of η^5 is much more accurate.
5. The t_2^2 configuration is indeed subject to a first-order Jahn-Teller distortion, and that is why the cluster is in fact D_{2d} -symmetric rather than perfectly T_d . Note that we only use the labels of the T_d point group to simplify the discussion. The distortion is apparent in the bond lengths of the Au_8 core, where the cube is distinctly flattened along one axis, and in fact the electronic configuration in the real structure is orbitally non-degenerate (e^2). This point has been noted in the text.
6. We thank the referee for pointing out the new $\text{M}@Pb_{12}$ structures, which are now included in the discussion.

REVIEWERS' COMMENTS:

Reviewer #2 (Remarks to the Author):

I am satisfied with the authors revisions and recommend publication in its present form.